# Geochemistry of Carboniferous–Permian Coal from the Wujiawan Mine, Datong Coalfield, Northern China: Modes of Occurrence, Origin of Valuable Trace Elements, and Potential Industrial Utilization

**Jialiang Ma [1,2]**, **Lin Xiao [1]**, **Ke Zhang [1]**, **Yukun Jiao [3]**, **Zhenzhen Wang [1]**, **Jinxiao Li [1]**, **Wenmu Guo [1]**, **Pengpeng Gao [1]**, **Shenjun Qin [1]** and **Cunliang Zhao [1,*]**

1   School of Earth Science and Engineering, Hebei University of Engineering, Handan 056038, China; Ma@iau.uni-frankfurt.de (J.M.); xiaolin@hebeu.edu.cn (L.X.); Zhangke9801@163.com (K.Z.); wzz90620512@163.com (Z.W.); lijinxiao1010@163.com (J.L.); gwm1992313@126.com (W.G.); gaopengpeng5693@163.com (P.G.); qinsj528@hebeu.edu.cn (S.Q.)
2   Institute for Atmospheric and Environmental Sciences, Faculty of Geoscience and Geography, Goethe-University Frankfurt, 60438 Frankfurt am Main, Germany
3   Hebei Provincal Coalfield Geology Bureau Geophysical Prospecting & Survey Administration, Xingtai 054000, China; jiaoyukun01@163.com
*   Correspondence: zhaocunliang@hebeu.edu.cn

**Abstract:** This paper provides new geochemical data focusing on valuable elements in the coal, parting, and floor samples in the No. 5 coal seam of the Taiyuan Formation from the Wujiawan mine, Datong coalfield, northern China. The minerals mainly consist of kaolinite, calcite, and pyrite, as well as trace amounts of quartz and illite. The No. 5 coal is enriched in Li, Ga, high field strength elements (HFSEs), and rare earth elements and yttrium (REY) when compared with world hard coals. Of particular interest is the high average concentration of Li (67.66 µg/g), which is around seven times higher than the value for world hard coals. Lithium, Ga, and HFSEs have strong inorganic affinities, whereas REY have organic affinities. The main carrier of Li, Ga, and HFSEs is aluminosilicate minerals, while REY appear to occur with organophosphorus. These HFSEs are enriched, both in the parting and in the adjacent coal samples. This suggests that these elements are likely to leach out during the diagenetic process. The distribution patterns of REY, along with the ratio of $Al_2O_3/TiO_2$ and the figure of $Zr/TiO_2$ vs. Nb/Y are suggestive of their derivation from felsic parent material. In the northern and eastern part of the Datong coalfield, there are several regions where the Li content is higher than the mineable grade, in particular in the northern Datong coalfield where there is a mine with an Li content of 294.6 µg/g. This is significantly higher than the mineable grade. Therefore, there is a potential for financially viable recovery of Li in these coals of the Datong coalfield.

**Keywords:** Datong coalfield; minerals; trace elements; felsic detrital; lithium

## 1. Introduction

According to the BP Statistical Review reported in 2019, global coal consumption increased by 1.4% in 2018—the fastest increase since 2013 [1]. Coal consumption in China has grown, and will continue to increase, due to rapid economic development [2]. The presence of trace elements in coal and their modes of occurrence are influenced by synsedimentary factors and modification during the peat formation period (e.g., biochemical and detrital input), diagenetic stage (e.g., detrital input and seawater), as well as during epigenetic stages (e.g., geological factors and hydrothermal solutions) [3–6].

Numerous investigations on the elemental geochemistry in coals from north China have been carried out [7–19] and have revealed that various valuable elements can be found in high concentrations which gives them considerable economic significance [9,15,17,20–23]. Dai et al. [24] suggested that rare metals, such as REY, Cr, Ti, Li, Be, Au, Ag, Pt, Pd, etc., have high potential for industrial utilization in some Chinese coals. Meanwhile, Dai and Finkelman [25] reported that the extraction and utilization of some strategically important elements from coal could result in economic benefits.

Datong coalfield is one of the most important coal-producing areas in China, and studies have been carried out on the mineralogy and geochemistry from the Carboniferous and Permian coals in, and adjacent to, the Datong coalfield [7,9,10,16–18,20,26–28]. Findings from these studies have shown that the sediment source is dominated by both Cambrian-Ordovician strata as well as Archean metamorphic rocks [9,10,16–18,20,26–29]. Additionally, the low temperature hydrothermal solution activities in this area have significantly influenced the abundance and modes of occurrence of trace elements present in the coal. Studies have indicated that the coals from Yanzishan mine in the Datong coalfield are slightly enriched in Al, Li, Ga, and Ge [29]. Moreover, it has proven possible to recover Al and Ge as valuable byproducts from coal ash [30,31]. Other studies have indicated that Li has both organic and inorganic affinities in coals [9,32,33]. According to Dai et al. [10,12], clay minerals, such as boehmite and kaolinite are the main carriers of Ga. However, some researchers have suggested that Ga is partly organically bound in coals [20,34]. The mineralogical composition of the coal in the Yongdingzhuang mine from the Datong coalfield includes kaolinite and quartz, with trace amounts of anatase and pyrite [35].

The majority of previous studies concentrated only on coals from the northern-central region of the Datong coalfield [29,35,36]. In contrast, there is limited research available on the coals from the southern area, such as the Wujiawan mine, one of the biggest mines in the Datong coalfield. In 2020, the geochemistry of the No.6 coal from the Nanyangpo mine [37], located 5 km from the Wujiawan mine, was investigated with a focus on the origin of potentially harmful elements. In this study, we investigated the abundance and modes of occurrence of trace elements, with a particular focus on enriched valuable elements in the No. 5 coal from the Wujiawan mine. The results provide new data on the enrichment of valuable elements in the coal from the south Datong coalfield. Furthermore, the detailed discussion on the modes of occurrence and enrichment of valuable elements should provide support for policy makers in China with regard to the comprehensive utilization of coal from the Datong coalfield.

## 2. Geological Background

The Datong coalfield is located in the north of Shanxi Province in northern China (see Figure 1). The Yinshang Oldland is the main source region for the coalfields of Shanxi, and include the Ningwu and Datong coalfields [36,38]. The Datong coalfield is about 50 km in length (north to south) and 30 km in width (west to east), with a total area of around 1900 km$^2$ [35]. The main coal-bearing strata in this basin are Carboniferous (C), Permian (P), and Jurassic (J). These coals in the Datong coalfield were deposited in a humid environment, and marine-terrigenous facies are the main deposits for C–P coals, whereas the J coals were mostly formed in a terrestrial lake environment. The Datong coalfield is bordered by the Lvliang mountain syncline, Pingwang-Emaokou fault, and the Hongtao mountain syncline to the east, west, and north, respectively. Coal-bearing sequences are present mainly in the Benxi, Taiyuan, Shanxi, Shihezi, and Shiqianfeng formations in the coalfield. Two stages of magmatic intrusions are reported in the Datong coalfield [37,39]: a lamprophyre intrusion that occurred during the middle–late Triassic and a diabase intrusion during the late Early Cretaceous [37,40]. Four reverse faults, including the Emaokou Fault, the Meiyukou Fault, the Wangjiayuan Fault, and the Qingciyao Fault, occur from south to north [29,35–37]. These faults may have provided the channels for the hydrothermal fluid to flow into the coal-bearing sequence [29,35,36].

The Wujiawan mine is situated in the southern region of the Datong coalfield with a total area of 9.37 km$^2$ and has an annual output of more than 3 million tons of coal. Borehole data indicate that the

strata of the study area are Cambrian, Ordovician, Carboniferous, Permian, Triassic, and Quaternary in ascending stratigraphic order. As Figure 2 shows, the Taiyuan Formation in the Carboniferous strata mainly consists of mudstone, siltstone, gritstone, pebbly-gritstone, carbonaceous-mudstone, and coal seams. The No.3, No. 5, No.8, and No.9 seams are the mineable coal seams of the Taiyuan Formation. The No. 5 coal seam has a total thickness of 33–138 m, with a floor mainly comprising of mudstone and a roof mainly of siltstone.

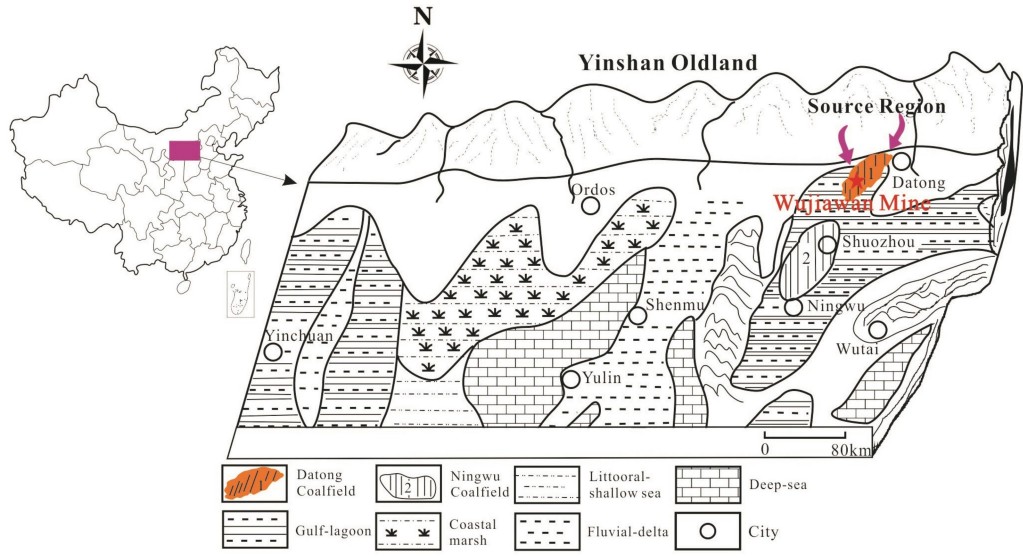

**Figure 1.** Paleogeographic setting and location of the Datong coalfield and Yinshan Oldland in the late Paleozoic. Modified from Sun et al. [9].

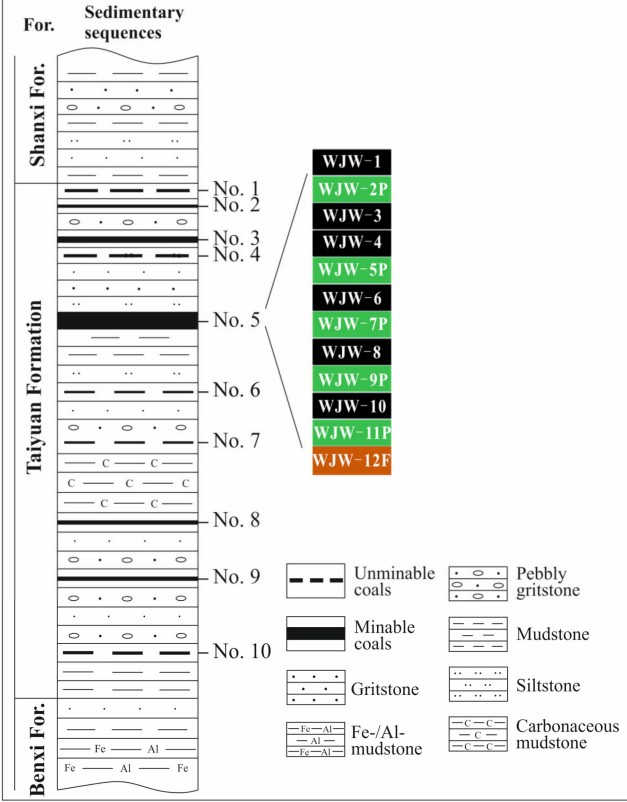

**Figure 2.** Stratigraphic column of the Wujiawan mine and lithological column of the sampling profile. For.: Formation, P: parting, F: floor.

## 3. Sampling and Methods

In the present study, 6 coal bench samples (WJW-1, WJW-3, WJW-4, WJW-6, WJW-8, and WJW-10) and 6 partings (WJW-2, WJW-5, WJW-7, WJW-9, WJW-11, WJW12) were collected from top to bottom of No. 5 coal from the Wujiawan mine, according to the method of GB/T 482-2008 [41]. The No. 5 coal was represented by a total thickness of 2.7 m, and every bench sample was cut with an area of 10 cm × 10 cm. All collected samples were split into two parts, one part for geochemical analyses and the other for future study.

Samples were crushed (by using a grinder whose material is hadifield steel for pulverization) to pass 200 mesh sieves for the proximate and related analyses. The analytical procedures of moisture, ash yield, volatile matter, and total sulfur were based on the ASTM Standards D3173-11 [42], D3174-11 [43], D3175-11 [44], and D3177-02 (2011) [45], respectively.

Samples were crushed into 80 mesh for optical microscopic and scanning electron microscopic analyses. According to ASTM D2798-05 [46], a Leica DM 2500 P (Leica, Solms, Germany) reflected light microscope coupled with a halogen lamp (Leica, Solms, Germany, oil lens 32/0.65, 548 nm, 3 × 3 μm, EMI9592 S-11; calibrated with a glass standard from Leitz, Ro = 0.89%) was used to measure random vitrinite reflectance (Ro, ran). Mineralogical occurrence and distribution were analyzed under optical microscope (Leica DM 2500 P microscope by Leica Microsystems, Solms, Germany) and scanning electron microscope in conjunction with an energy dispersive X-ray spectrometer (SEM-EDS, Hitachi SU8220, Tokyo, Japan). For SEM-EDS, each sample was platinum-plate coated (Although samples for SEM were platinum-plate coated, sometimes it could not be detected by EDS due to the very low concentration, especially in the organic area (on macerals)). All the samples were subjected to low temperature ashing (LTA), then the LTA samples, as well as powdered parting and floor samples, were detected under X-ray powder diffraction (XRD). The detailed analytical procedures of XRD are described in Ma et al. [37].

The 200-mesh samples were ashed at 815 °C and the resultant ashes were then studied under X-ray fluorescence spectrometry (XRF) (ARL9800 XRF, Thermo Fisher Scientific, Waltham, MA, USA) to analyze the proportion of major elements. Meanwhile, the loss on ignition was also calculated at this temperature. The matrix correction and calibration of XRF is described by Ma et al. [37]. The contents of trace elements in the parting, floor, and coal samples were measured by Xseries II ICP-MS (Thermo Fisher Scientific, Waltham, MA, USA). Prior to the ICP-MS analysis, an amount of 40 mg sample (<200 mesh) was weighed into Poly Tetra Fluoro Ethylene (PTFE) vessels for microwave digestion, 2 mL of HF (50%) + 5 mL of HNO₃ (65%) + 2 mL of H₂O₂ (30%) were added, and microwave digestion was performed for 75 min at a temperature of 200 °C. This solution was then transferred into 125-mL fluorinated ethylene propylene (FEP) bottles that were filled with 100 g of deionized water.

## 4. Results

### 4.1. Bulk Coal Characteristics

Table 1 shows the results of the proximate and related analyses of the 6 coal samples from the Wujiawan mine. Ash yields from the Wujiawan coal samples ranged from 13.60% to 31.15%, with a mean ash yield of 18.72%. Consequently, the coal seam was classified as low-ash coal, in accordance with Chinese Standards GB/T 15224.1-2010 [47]. The volatile matter content of the No. 5 coal samples ranged from 24.76% to 34.72%, with an average of 30.13%, suggesting that the No. 5 coal is medium–high-volatility coal in accordance with the standards of the MT/T 849-2000(28.01–37.00% for medium–high-volatility coal) [48]. The moisture content ranged from 3.29% to 5.38%, averaging 4.21%, and qualifies as low moisture coal based on the MT/T 850–2000 (<5% for low moisture coal) [49]. Total sulfur content for the No. 5 coal ranged from 0.50% to 3.89%, with a mean of 1.35%. These samples are considered to be medium-sulfur-coal, in accordance with Chinese Standards GB/T 15224.2-2010 [50], which classifies coals with a total sulfur content 1.01%–2.00% as medium-sulfur-coal. The average

vitrinite reflectance was found to be 0.61% (ranging from 0.57%–0.63%), indicating a bituminous coal, according to the ASTM Standard [46].

**Table 1.** Proximate analysis and the reflectance of the No. 5 coal seam from the Wujiawan mine.

| Sample | $A_d$ | $V_{daf}$ | $M_{ad}$ | $S_{t,d}$ | $R_{o, ran}$ |
|--------|-------|-----------|----------|-----------|--------------|
| WJW5-1 | 17.85 | 30.46 | 4.14 | 0.63 | 0.63 |
| WJW5-3 | 22.15 | 32.68 | 4.47 | 0.79 | 0.57 |
| WJW5-4 | 31.15 | 24.76 | 3.29 | 0.50 | 0.59 |
| WJW5-6 | 13.60 | 32.70 | 4.01 | n.a. | 0.61 |
| WJW5-8 | 14.15 | 25.46 | 3.97 | 0.93 | 0.62 |
| WJW5-10 | 13.40 | 34.72 | 5.38 | 3.89 | 0.62 |
| AVE | 18.72 | 30.13 | 4.21 | 1.35 | 0.61 |

A, ash yield; V, volatile matter; M, moisture; $S_t$, total sulfur; $R_{o, ran}$, random reflectance of vitrinite; d, dry basis; daf, dry and ash-free basis; ar, as-received basis; AVE, average concentration of the Wujiawan coal samples; n.a., not analyzed.

The No. 5 coals from the Wujiawan mine are classified as low-ash, medium–high-volatility, low moisture, medium-sulfur, bituminous coals.

## 4.2. Mineralogical Characteristics

Based on the results of XRD analysis of the LTA, optical microscopy, and SEM-EDS (see Figures 3 and 4), the No. 5 coal samples from the Wujiawan mine mainly contain kaolinite, calcite, and pyrite. In addition, trace amounts of quartz and illite were found by the SEM-EDS in the coal samples.

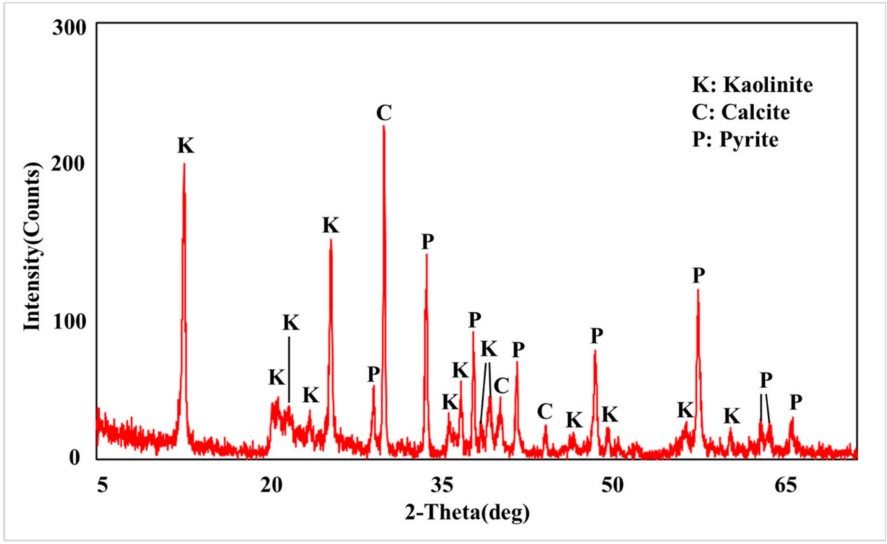

**Figure 3.** The XRD pattern of LTA residues of sample WJW-6 from Wujiawan mine, Datong coalfield.

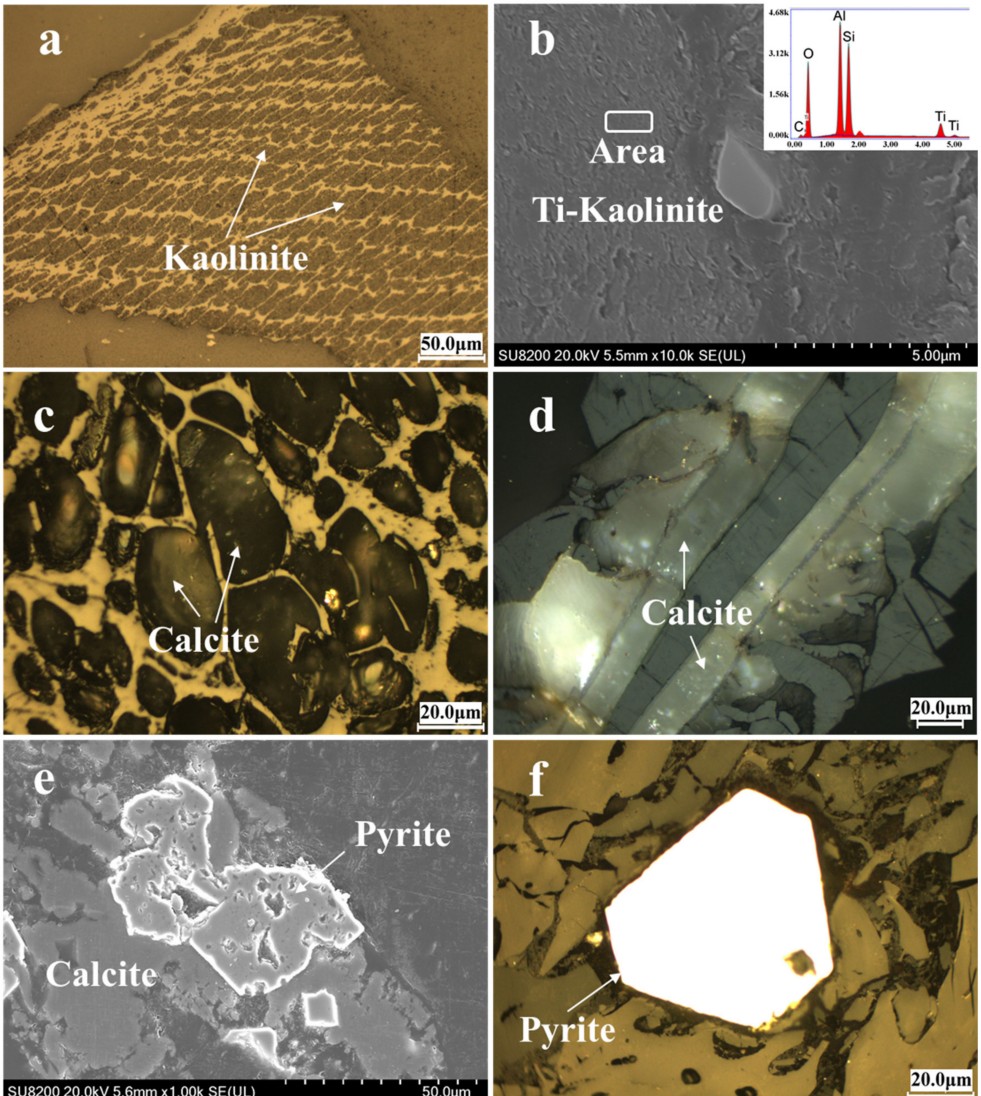

**Figure 4.** Distribution of the main minerals in the No. 5 coals from the Wujiawan mine, Datong coalfield. (**a**) Cell-filling kaolinite (reflected light); (**b**) fragmental Ti-bearing kaolinite and laminated kaolinite (secondary electron excitation); (**c**) cell-filling calcite (reflected light, oil immersion); (**d**) fracture-filling calcite (reflected light, oil immersion); (**e**) pyrite co- occurred with calcite (secondary electron excitation); and (**f**) pyrite crystal (reflected light, oil immersion).

### 4.2.1. Kaolinite

XRD, SEM-EDS, and optical microscopy data showed that kaolinite is the main mineral in the Wujiawan coal and mainly occurs as cell-filling (see Figure 4a). There were also trace amounts of Ti-bearing kaolinite, detected by SEM-EDS, (see Figure 4b). The cell-filling kaolinite (see Figure 4a) is common in coal [21,36,51–53] and is regarded to have both authigenic and syndepositional origin [54,55]. Furthermore, the authigenic kaolinite may be precipitated from an acidic solution that favors the formation of authigenic kaolinite [56]. Ti-bearing kaolinite has also been reported in the late Permian coals seams located at the Songzao coalfield, in southwestern China [57]. The cell-filling Ti-bearing kaolinite indicates that Ti is associated with kaolinite during the authigenic process.

### 4.2.2. Calcite

Calcite is frequently formed during the authigenic, epigenetic, and postcoalification stages [58]. Various forms of calcite in the No. 5 coals from the Wujiawan mine were observed under optical

microscopy and SEM-EDS (e.g., cell-filling (see Figure 4c), fracture-filling (see Figure 4d), and calcite-pyrite aggregates (see Figure 4e, etc.). The cell- and fracture-filling calcite indicate an authigenic origin [59,60].

### 4.2.3. Pyrite

Pyrite is one of the most common minerals in coals [61,62]. The pyrite in the Wujiawan coal mostly occurs in crystal form (see Figure 4f) and co-fracture-filling with calcite (see Figure 4e). As Vejahati et al. [63] reported, the crystals of pyrite formed during the early diagenesis, whereas fissure-filled pyrite formed during the epigenetic stage. The different occurrences of pyrite indicate that the pyrite in the Wujiawan coals has both syngenetic and epigenetic origin.

### *4.3. Geochemistry*

### 4.3.1. Major Element Oxides

A comparison of the proportion of major oxides in the No. 5 coal samples from the Wujiawan with the mean values of Chinese coals [64] is shown in Table 2. The data show that the $SiO_2/Al_2O_3$ ratio of the No. 5 coal (1.05) and partings (1.13) are lower than the common values of Chinese coals (1.42) [64], despite the fact that oxides of the major elements in the Wujiawan coal are dominated by $SiO_2$ (8.50%) and $Al_2O_3$ (8.07%) (Table 2). These values are very close to the theoretical value of kaolinite (1.18). This can be attributed to the low quartz content in the Wujiawan coal. In addition, the No. 5 coal samples are also slightly enriched in $P_2O_5$ and MnO. The remaining major element oxides (CaO, $TiO_2$, $Fe_2O_3$, MgO, $K_2O$, and $Na_2O$) have values lower than the corresponding average values in Chinese coals.

**Table 2.** Proportion of major element oxides in the Wujiawan coal samples and their comparisons with the mean value of Chinese coals (Values are in wt% on a whole coal basis).

| Sample | $SiO_2$ | $Al_2O_3$ | CaO | $P_2O_5$ | $TiO_2$ | $Fe_2O_3$ | MgO | $K_2O$ | $Na_2O$ | MnO | $SiO_2/Al_2O_3$ |
|---|---|---|---|---|---|---|---|---|---|---|---|
| WJW-5-1 | 8.03 | 8.08 | 0.54 | 0.49 | 0.13 | 0.13 | 0.05 | 0.01 | 0.01 | 0.002 | 0.99 |
| WJW-5-2P | 37.8 | 34.93 | 0.06 | 0.03 | 0.67 | 0.32 | 0.14 | 0.1 | 0.03 | bdl | 1.08 |
| WJW-5-3 | 9.68 | 9.2 | 2.02 | 0.03 | 0.13 | 0.27 | 0.11 | 0.04 | 0.01 | 0.011 | 1.05 |
| WJW-5-4 | 14.79 | 13.7 | 0.82 | 0.04 | 0.44 | 0.27 | 0.13 | 0.07 | 0.04 | 0.005 | 1.08 |
| WJW-5-5P | 34.69 | 31.2 | 0.09 | 0.02 | 0.39 | 0.18 | 0.15 | 0.25 | 0.05 | bdl | 1.11 |
| WJW-5-6 | 5.84 | 5.65 | 1.09 | 0.01 | 0.1 | 0.96 | 0.05 | 0.01 | 0.01 | 0.007 | 1.03 |
| WJW-5-7P | 27.37 | 24.75 | 0.1 | 0.04 | 0.89 | 0.17 | 0.12 | 0.1 | 0.03 | bdl | 1.11 |
| WJW-5-8 | 6.47 | 6.05 | 0.88 | 0.01 | 0.13 | 0.2 | 0.05 | 0.01 | 0.01 | 0.004 | 1.07 |
| WJW-5-9P | 33.14 | 29.82 | 0.14 | 0.01 | 0.32 | 0.18 | 0.15 | 0.11 | 0.02 | bdl | 1.11 |
| WJW-5-10 | 6.21 | 5.75 | 0.16 | 0.01 | 0.18 | 0.62 | 0.06 | 0.01 | 0.01 | 0.066 | 1.08 |
| WJW-5-11P | 40.5 | 31.9 | 0.21 | 0.05 | 0.93 | 0.91 | 0.36 | 0.85 | 0.05 | 0.002 | 1.27 |
| WJW-5-12F | 35.47 | 32.76 | 0.08 | 0.02 | 1.13 | 0.21 | 0.12 | 0.07 | 0.03 | bdl | 1.08 |
| AVE-C | 8.50 | 8.07 | 0.92 | 0.10 | 0.19 | 0.41 | 0.08 | 0.03 | 0.02 | 0.02 | 1.05 |
| AVE-P | 34.83 | 30.89 | 0.11 | 0.03 | 0.72 | 0.33 | 0.17 | 0.25 | 0.04 | bdl | 1.13 |
| *China | 8.47 | 5.98 | 1.23 | 0.092 | 0.33 | 4.85 | 0.22 | 0.19 | 0.16 | 0.015 | 1.42 |

AVE-C: average of coal samples; AVE-P: average of parting samples; bdl: below detection limit; and *China: average content of major-element oxides for common Chinese coals are from Dai et al. [64].

### 4.3.2. Trace Elements

Table 3 lists the concentration of trace elements in the No. 5 coal from the Wujiawan mine and their comparison with other world hard coals [65]. The results indicate that Li is highly enriched in the No. 5 coal at the Wujiawan mine, with a concentration coefficient (CC, the ratio of element concentration in Wujiawan coal and the corresponding average value in world hard coals) >5 (see Figure 5a). Gallium, Zr, Nb, Hf, Ta, and Th are enriched (2< CC < 5) (see Figure 5a), while some elements, such as Be, Sc, Cd, W, Pb, and U (0.5 < CC < 2), are at levels near to average values of world hard coals (see Figure 5a). The remaining elements are underrepresented in the No. 5 coals (see Figure 5a).

**Table 3.** Contents of trace elements in the No. 5 coal from the Wujiawan mine, Datong coalfield (µg/g, on whole coal basis).

| Samples | Li | Be | Sc | V | Cr | Co | Ni | Cu | Zn | Ga | Rb | Zr | Nb | Mo | Cd | Cs | Ba | Hf | Ta | W | Pb | Bi | Th | U |
|---|---|---|---|---|---|---|---|---|---|---|---|---|---|---|---|---|---|---|---|---|---|---|---|---|
| WJW-5-1 | 97.4 | 3.17 | 6.8 | 18.3 | 4.6 | 0.5 | 1.8 | 10.7 | 6.6 | 16.6 | 0.4 | 140.1 | 7.06 | 0.57 | 0.08 | 0.06 | 63.0 | 3.71 | 0.36 | 0.32 | 15.6 | 0.33 | 7.29 | 2.90 |
| WJW-5-2P | 411.6 | 1.46 | 7.4 | 22.8 | 7.3 | 0.4 | 2.1 | 14.2 | 11.0 | 32.7 | 3.6 | 333.0 | 33.71 | 1.08 | 0.25 | 0.43 | 18.1 | 10.10 | 2.57 | 2.31 | 14.0 | 0.89 | 22.69 | 4.38 |
| WJW-5-3 | 77.5 | 2.09 | 4.9 | 12.2 | 5.1 | 0.7 | 2.1 | 3.5 | 8.7 | 27.4 | 1.7 | 108.1 | 12.50 | 0.97 | 0.11 | 0.18 | 18.7 | 3.81 | 0.63 | 0.50 | 12.7 | 0.46 | 8.90 | 4.03 |
| WJW-5-4 | 132.9 | 3.34 | 7.5 | 17.1 | 8.1 | 0.3 | 1.7 | 10.6 | 9.4 | 19.0 | 2.3 | 163.4 | 15.46 | 0.89 | 0.16 | 0.29 | 23.1 | 5.62 | 1.15 | 1.61 | 27.5 | 0.66 | 17.95 | 3.42 |
| WJW-5-5P | 143.1 | 1.06 | 3.9 | 12.1 | 4.3 | 0.3 | 0.6 | 6.9 | 6.0 | 28.5 | 9.9 | 123.9 | 19.44 | 1.33 | 0.14 | 0.77 | 23.7 | 4.41 | 1.68 | 1.91 | 6.5 | 0.25 | 12.02 | 2.02 |
| WJW-5-6 | 31.3 | 2.53 | 3.7 | 8.6 | 2.5 | 0.6 | 1.2 | 3.5 | 7.2 | 8.7 | 0.3 | 71.1 | 3.56 | 0.72 | <0.05 | 0.03 | 11.8 | 2.37 | 0.31 | 0.83 | 10.5 | 0.12 | 8.59 | 2.14 |
| WJW-5-7P | 181.3 | 1.11 | 4.6 | 11.8 | 3.4 | 0.2 | 1.4 | 11.8 | 13.2 | 24.7 | 3.5 | 188.6 | 24.98 | 1.85 | 0.23 | 0.32 | 12.8 | 5.81 | 2.57 | 6.15 | 9.7 | 0.93 | 26.42 | 4.36 |
| WJW-5-8 | 44.1 | 2.56 | 3.1 | 7.3 | 1.7 | 0.4 | 1.2 | 3.8 | 5.5 | 16.4 | 0.3 | 112.5 | 8.73 | 1.27 | <0.05 | 0.03 | 7.0 | 3.54 | 0.58 | 0.94 | 9.3 | 0.18 | 9.71 | 2.84 |
| WJW-5-9P | 132.0 | 1.40 | 3.7 | 5.8 | 1.6 | 0.2 | 0.5 | 3.1 | 7.0 | 29.3 | 3.8 | 143.1 | 16.37 | 0.94 | <0.05 | 0.29 | 18.2 | 4.68 | 1.27 | 1.24 | 10.7 | 0.19 | 4.81 | 1.07 |
| WJW-5-10 | 22.7 | 3.51 | 3.4 | 11.0 | 3.6 | 0.6 | 2.9 | 2.6 | 4.3 | 18.2 | 0.2 | 123.5 | 10.01 | 1.73 | <0.05 | 0.03 | 2.9 | 3.70 | 0.36 | 0.44 | 11.5 | 0.13 | 5.77 | 3.29 |
| WJW-5-11P | 195.4 | 3.18 | 16.6 | 132.3 | 115.5 | 7.2 | 25.3 | 22.7 | 41.4 | 34.8 | 34.7 | 205.4 | 18.72 | 0.86 | 0.14 | 3.87 | 110.8 | 6.07 | 1.07 | 2.26 | 21.7 | 0.44 | 16.89 | 4.41 |
| WJW-5-12F | 345.9 | 1.33 | 9.4 | 36.7 | 8.9 | 0.5 | 3.0 | 19.3 | 21.7 | 32.5 | 2.0 | 472.9 | 39.70 | 4.32 | 0.38 | 0.23 | 17.7 | 10.59 | 2.82 | 4.05 | 14.3 | 1.04 | 20.62 | 5.44 |
| AVE-C | 67.66 | 2.87 | 4.90 | 12.42 | 4.29 | 0.52 | 1.81 | 5.77 | 6.95 | 17.72 | 0.88 | 119.78 | 9.56 | 1.03 | 0.11 | 0.10 | 21.09 | 3.79 | 0.56 | 0.77 | 14.53 | 0.31 | 9.70 | 3.10 |
| AVE-P | 234.88 | 1.59 | 7.60 | 36.92 | 23.50 | 1.47 | 5.48 | 13.00 | 16.72 | 30.42 | 9.58 | 244.48 | 25.49 | 1.73 | 0.19 | 0.99 | 33.55 | 6.94 | 2.00 | 2.99 | 12.82 | 0.62 | 17.24 | 3.61 |
| world coals [1] | 10 | 1.6 | 3.90 | 28.00 | 16 | 5.1 | 13 | 16 | 23 | 5.8 | 8.3 | 36.00 | 3.70 | 2.20 | 0.22 | 1 | 150 | 1.20 | 0.28 | 1.10 | 7.8 | 0.97 | 3.30 | 2.4 |
| world clays [2] | 54 | 3 | 15 | 120 | 110 | 19 | 49 | 36 | 89 | 16 | 133 | 190 | 11 | 1.6 | 0.91 | 13 | 460 | 120 | 110 | 2.6 | 14 | 0.38 | 4.3 | 4.3 |

[1] World coals, average content of trace elements for world coals are from Ketris and Yudovich [65]; [2] world clays, average content of trace elements for world clays are from Yaroshevsky [66]; AVE-C, average content of coal samples; and AVE-P, average content of parting samples.

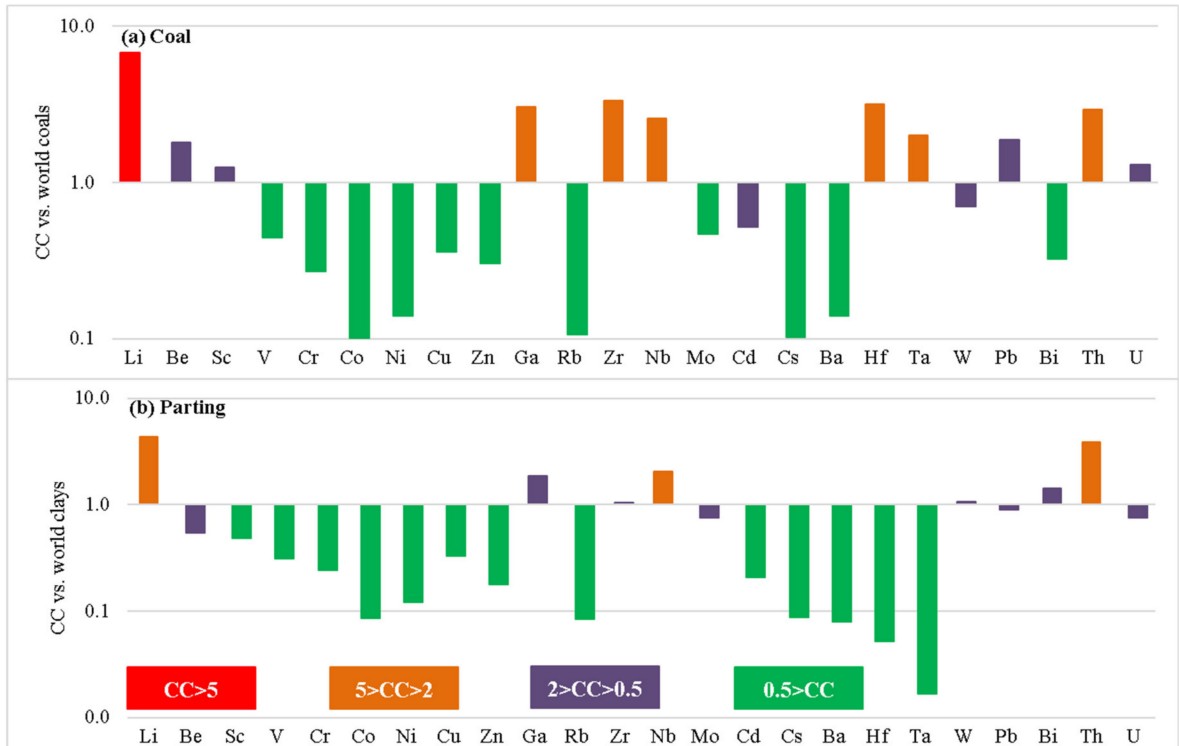

**Figure 5.** Concentration coefficients of trace elements in coals (**a**) and partings (**b**) from the Wujiawan mine at Datong coalfield, normalized by the average concentrations of trace elements in world hard coals and world clays, respectively.

Lithium, Nb, and Th in the partings of the No. 5 coal seam are enriched (see Figure 5b) compared to world clays [66]. Values for Be, Ga, Zr, Mo, W, Pb, Bi, and U approach the mean values for world clays, and the remaining elements are depleted when compared to world clays.

### 4.3.3. Rare Earth Elements and Yttrium (REY)

The concentration of REY in the Wujiawan coal varied from 45.14 to 372.60 µg/g, with a mean of 157.86 µg/g (Table 4). This is around 2.5 times higher than the average value of world hard coals (68.41 µg/g). The average content of REY in the parting samples (151.22 µg/g) was slightly lower than that in the coal samples (Table 4). The REY content in coal samples WJW-1 (372.60 µg/g) and WJW-4 (194.09 µg/g) was much higher than for other coal samples. Furthermore, the REY content in the parting samples WJW-7 and WJW-11 were 393.48 and 220.34 µg/g, respectively, and were the highest two samples in the No. 5 coal seam.

In the present study, three types of REY—light (LREY: La, Ce, Pr, Nd, and Sm), medium (MREY: Eu, Gd, Tb, Dy, and Y), and heavy (HREY: Ho, Er, Tm, Yb, and Lu)—were identified in accordance with the Seredin and Dai's classification parameters [4]. The concentration of REY was normalized by the UCC to classify the enrichment types of REY distribution. Seredin and Dai [67] identify the enrichment types of REY as L-type (light-REY: $La_N/Lu_N > 1$), M-type (medium-REY: $La_N/Sm_N < 1$, $Gd_N/Lu_N > 1$), and H-type (heavy REY: $La_N/Lu_N < 1$) in this coal. At the same time, a mixed type (H-M-type) also occurs in the No. 5 coal. The Eu anomalies ($Eu_N/Eu_N^*$) and Gd anomalies ($Gd_N/Gd_N^*$) were calculated according to formulae 4 and 5 from Dai et al. [68].

**Table 4.** Concentrations of Rare Earth Elements and Y (REY) in the coals and partings from the No. 5 coal seam.

| Sample | WJW5-1 | WJW5-2P | WJW5-3 | WJW5-4 | WJW5-5P | WJW5-6 | WJW5-7P | WJW5-8 | WJW5-9P | WJW5-10 | WJW5-11P | WJW-12F | AVE-C | AVE-P |
|---|---|---|---|---|---|---|---|---|---|---|---|---|---|---|
| La | 107.82 | 12.19 | 18.63 | 36.12 | 18.71 | 16.74 | 92.11 | 18.08 | 7.92 | 9.59 | 47.45 | 5.26 | 34.50 | 30.61 |
| Ce | 151.46 | 30.02 | 40.89 | 73.38 | 36.36 | 32.48 | 164.69 | 35.84 | 14.92 | 20.34 | 84.29 | 17.82 | 59.06 | 58.02 |
| Pr | 15.60 | 3.34 | 4.69 | 7.95 | 3.70 | 3.58 | 18.06 | 3.94 | 1.52 | 2.27 | 9.12 | 2.27 | 6.34 | 6.33 |
| Nd | 51.86 | 11.38 | 17.36 | 28.23 | 12.31 | 12.88 | 59.57 | 13.69 | 4.90 | 8.14 | 30.50 | 8.44 | 22.03 | 21.18 |
| Sm | 8.73 | 2.47 | 3.86 | 5.74 | 2.12 | 2.71 | 10.32 | 2.72 | 0.85 | 1.84 | 5.27 | 2.31 | 4.27 | 3.89 |
| Eu | 1.42 | 0.45 | 0.67 | 0.97 | 0.34 | 0.50 | 1.67 | 0.54 | 0.17 | 0.45 | 1.31 | 0.57 | 0.76 | 0.75 |
| Gd | 8.94 | 2.57 | 3.82 | 5.74 | 2.09 | 2.72 | 10.05 | 2.80 | 0.99 | 1.90 | 5.50 | 2.67 | 4.32 | 3.98 |
| Tb | 0.89 | 0.54 | 0.71 | 0.90 | 0.26 | 0.47 | 1.14 | 0.42 | 0.17 | 0.38 | 0.79 | 0.70 | 0.63 | 0.60 |
| Dy | 4.04 | 3.39 | 4.15 | 4.75 | 1.51 | 2.61 | 5.15 | 2.51 | 1.30 | 2.24 | 4.64 | 4.24 | 3.38 | 3.37 |
| Y | 16.71 | 17.83 | 21.81 | 23.20 | 8.30 | 14.78 | 23.55 | 14.57 | 9.51 | 13.66 | 23.98 | 20.54 | 17.45 | 17.29 |
| Ho | 0.76 | 0.75 | 0.90 | 1.00 | 0.34 | 0.58 | 1.01 | 0.57 | 0.35 | 0.51 | 1.04 | 0.92 | 0.72 | 0.73 |
| Er | 2.03 | 1.97 | 2.38 | 2.65 | 0.93 | 1.57 | 2.76 | 1.57 | 1.06 | 1.43 | 2.81 | 2.31 | 1.94 | 1.97 |
| Tm | 0.28 | 0.31 | 0.38 | 0.43 | 0.15 | 0.25 | 0.41 | 0.26 | 0.19 | 0.24 | 0.46 | 0.37 | 0.31 | 0.31 |
| Yb | 1.79 | 1.81 | 2.28 | 2.67 | 0.89 | 1.49 | 2.60 | 1.56 | 1.13 | 1.51 | 2.79 | 2.23 | 1.89 | 1.91 |
| Lu | 0.26 | 0.27 | 0.34 | 0.39 | 0.13 | 0.23 | 0.38 | 0.24 | 0.17 | 0.22 | 0.41 | 0.32 | 0.28 | 0.28 |
| REY | 372.60 | 89.29 | 122.86 | 194.09 | 88.14 | 93.58 | 393.48 | 99.30 | 45.14 | 64.72 | 220.34 | 70.94 | 157.86 | 151.22 |
| $La_N/Lu_N$ | 4.41 | 0.49 | 0.59 | 0.99 | 1.49 | 0.79 | 2.56 | 0.80 | 0.51 | 0.47 | 1.24 | 0.18 | 1.34 | 1.08 |
| $La_N/Sm_N$ | 1.85 | 0.74 | 0.72 | 0.94 | 1.33 | 0.93 | 1.34 | 1.00 | 1.40 | 0.78 | 1.35 | 0.34 | 1.04 | 1.08 |
| $Gd_N/Lu_N$ | 2.89 | 0.82 | 0.95 | 1.25 | 1.31 | 1.02 | 2.20 | 0.98 | 0.50 | 0.73 | 1.14 | 0.70 | 1.30 | 1.11 |
| $Eu_N/Eu_N*$ | 0.91 | 0.80 | 0.81 | 0.83 | 0.87 | 0.88 | 0.89 | 0.98 | 0.93 | 1.08 | 1.25 | 0.92 | 0.92 | 0.94 |
| $Gd_N/Gd_N*$ | 1.49 | 0.90 | 0.98 | 1.11 | 1.29 | 1.03 | 1.35 | 1.15 | 1.09 | 0.94 | 1.19 | 0.78 | 1.12 | 1.10 |
| Type | L-type | H-type | H-type | H-M-type | L-type | H-M-type | L-type | H-type | H-type | H-type | L-type | H-type | - | - |

AVE-C: average of coal samples; AVE-P: average of parting samples; REY: the sum of rare earth elements and yttrium; subscript N indicates values are normalized by the average content of the Upper Continental Crust (UCC) [4,5]; $Eu_N/Eu_N* = Eu_N/(0.67Sm_N + 0.33Tb_N)$; and $Gd_N/Gd_N* = Gd_N/(0.33Sm_N + 0.67Tb_N)$ [68].

The REY enrichment patterns in No. 5 coals (including partings) were mainly characterized by H-type (see Figure 6a) and L-type enrichment (see Figure 6b). Only the WJW-4 and WJW-6 samples were found to be of the M-H-REY type (see Figure 6c).

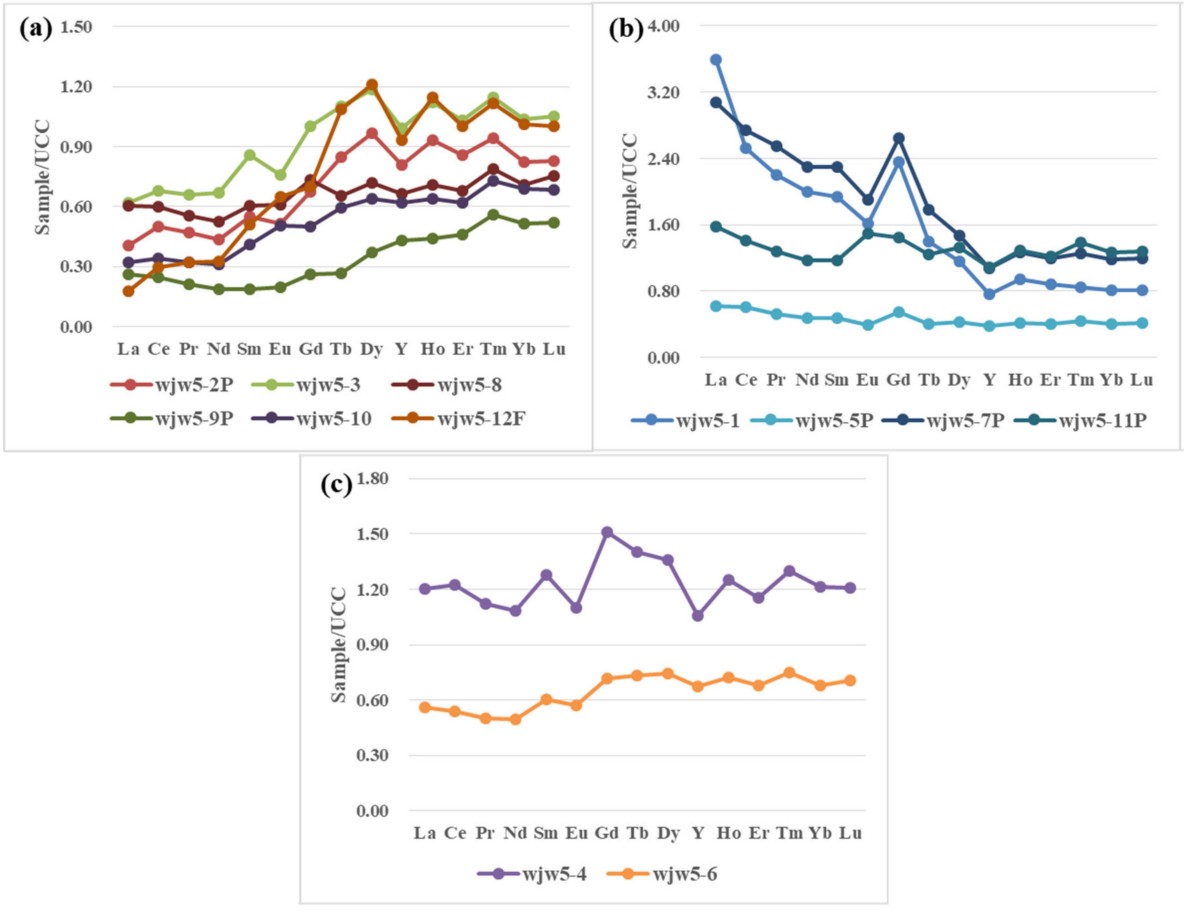

**Figure 6.** REY distribution patterns in the No. 5 coal (including coal and parting samples), normalized to the upper continental crust. (**a**) Patterns of H-type enrichment; (**b**), patterns of L-type enrichment; and (**c**), patterns of M-H-REY enrichment.

## 5. Discussion

### 5.1. Affinity of the Elements

The elements' affinity in coal for organic or inorganic matter may be reflected in the correlation between the element concentrations and ash yield [69,70]. Numerous studies have discussed the characteristics of elements in coal, e.g., occurrence, affinity, and genesis, by using correlation analysis [71–73]. However, the analysis method of correlation analysis should also be noted, because element correlations are dependent on the homogeneity of the population, the number of samples, and the elements determined. A good selection of samples would decrease the possibility that the statistical manipulation of the data would suggest an impossible element relationships [70,74]. Figure 7a,b shows the correlations (r) of ash yield with major element oxides, and trace elements, respectively.

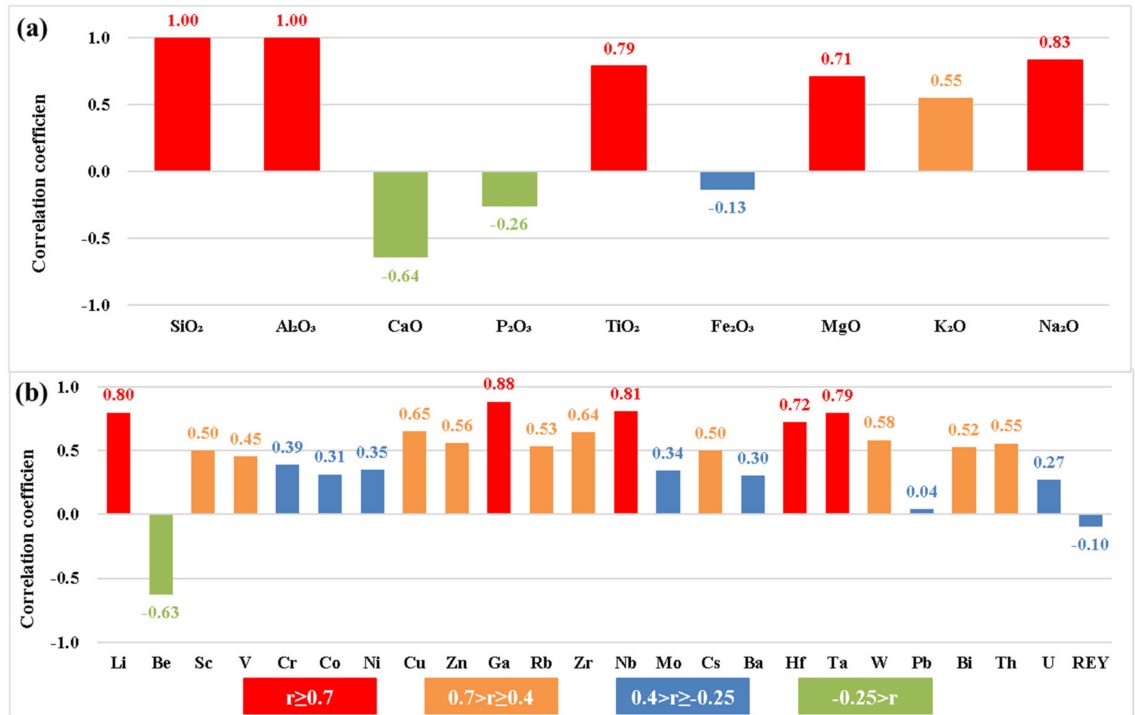

**Figure 7.** Correlation coefficients between ash yield and elements in Wujiawan coal. (**a**) The correlation coefficients between ash yield and major elements; (**b**) the correlation coefficients between ash yield and trace elements.

Four groups (Groups 1–4) of elements have been recognized, based on the relationship between their ash yield and elemental concentrations:

Group 1 consists of $Al_2O_3$, $SiO_2$, $TiO_2$, MgO, $Na_2O$, Li, Ga, Nb, Hf, and Ta. The concentrations of elements from this group are significantly correlated with the ash yield, suggesting a high inorganic affinity. Furthermore, the strong affinity between $Al_2O_3$ and $SiO_2$ (0.99, Table 5) suggests that the kaolinite in the samples is the most important carrier of $Al_2O_3$ and $SiO_2$. This is supported by the high content of kaolinite and simultaneously low quartz content, as obtained from the results of the optical microscopy, XRD, and SEM-EDS. Additionally, the $TiO_2$ in the No. 5 coal has strong and similar correlations with $Al_2O_3$ and $SiO_2$ (r = 0.79 and 0.78, respectively), showing that kaolinite is also the main carrier of Ti. This is also supported by the EDS-detectable Ti within kaolinite under SEM-EDS, as well as the low content of titanium oxide minerals such as rutile, brookite, and anatase in this coal. The Ti-bearing kaolinite in coal from the Songzao coalfield and Hunchun coalfield, in China, has been reported by Zhao et al. [19] and Dai et al. [75], respectively. Both investigations suggested that Ti occurs as submicron oxides in kaolinite. The remaining elements in this group (MgO, $Na_2O$, Li, Ga, Nb, Hf, and Ta) also have high correlations with $Al_2O_3$ and $SiO_2$ ($r_{(Al, Si)} > 0.7$, Table 5), along with an absence of accessory minerals (e.g., zircon), indicating an aluminosilicate affinity mainly associated with clay minerals.

Group 2 includes the following elements: $K_2O$, Sc, V, Cu, Zn, Rb, Zr, Cs, W, Bi, and Th. This group has a relatively low inorganic affinity compared to Group 1, but relatively higher r values than other groups ($r_{ash}$ = 0.4–0.69, Figure 7a,b). Furthermore, elements in this group have moderate correlations with $SiO_2$ and $Al_2O_3$ ($r_{(Al, Si)}$ = 0.4–0.69, Table 5), indicating a moderate aluminosilicate mineral affinity.

Group 3 consists of $Fe_2O_3$, Cr, Co, Ni, Mo, Ba, Pb, U, and REY. The r values of the elements in this group vary from −0.13 to 0.39 (see Figure 7a and b), suggesting either organic or inorganic affinities. Among these elements, Cr has a relatively high correlation coefficient with ash yield (r = 0.39, see Figure 7b), suggesting that Cr has probably more affinity for inorganic matter; this is further supported by relatively high correlation with $SiO_2$ (0.44) and $Al_2O_3$ (0.34, Table 5).

**Table 5.** Correlation coefficients between the concentrations of trace and major elements in the Wujiawan coal and selected elements.

| Aluminosilicate Affinity | | | |
|---|---|---|---|
| $r_{(Al,\,Si)} > 0.7$ | Li (0.78, 0.82) Ta (0.77, 0.82) | Ga (0.88, 0.87) $TiO_2$ (0.79, 0.78) | Nb (0.79, 0.82) MgO (0.74, 0.67) | Hf (0.71, 0.74) $Na_2O$ (0.84, 0.83) |
| $r_{(Al,\,Si)} = 0.4$–0.69 | $K_2O$ (0.60, 0.50) Cu (0.67, 0.64) Cs (0.55, 0.46) | Sc (0.53, 0.47) Zn (0.59, 0.52) W (0.58, 0.59) | V (0.50, 0.41) Rb (0.58, 0.49) Bi (0.51, 0.54) | Cr (0.44, 0.34) Zr (0.62, 0.66) Th (0.55, 0.57) |

| Correlation Coefficients between Selected Elements | | | | | |
|---|---|---|---|---|---|
| $Al_2O_3$–$SiO_2$ = 0.99 | CaO–$SiO_2$ = −0.65 | CaO–$Al_2O_3$ = −0.66 | Li–Zr (0.90) | Li–Nb (0.93) | Li–Hf (0.97) |
| Li–Ta (0.87) | Li–Bi (0.82) | Ga–$TiO_2$ (0.70) | Zr–Nb (0.92) | Zr–Hf (0.96) | Zr–Ta (0.80) |
| Zr–Th (0.65) | Nb–Ta (0.95) | Nb–Hf (0.96) | Nb–Th (0.76) | Hf–Ta (0.86) | Hf–Th (0.74) |
| Ta–Th (0.83) | $TiO_2$–Zr (0.81) | $TiO_2$–Nb (0.86) | $TiO_2$–Hf (0.82) | $TiO_2$–Ta (0.83) | $TiO_2$–Th (0.83) |

Group 4 only includes CaO, $P_2O_5$, and Be; the correlation coefficient values between them and ash yield range from −0.64 to −0.26. The relatively low correlation coefficient of these elements with ash yield showing an organic affinity.

## 5.2. Sediment Source Region

The ratio of $Al_2O_3$/$TiO_2$ could be an indicator of the source region for the coal deposits [76], since it may remain almost unchanged during surface weathering and alteration [77–79]. Generally, the value of $Al_2O_3$/$TiO_2$ for sediments derived from mafic is 3–8, from intermediate is 8–21, and from felsic rocks is 21–70 [5]. Figure 8a shows that the ratios of $Al_2O_3$/$TiO_2$ for all samples in the No. 5 coal (including parting and floor samples) are higher than 21 (see Figure 8a), suggesting a felsic sediment source.

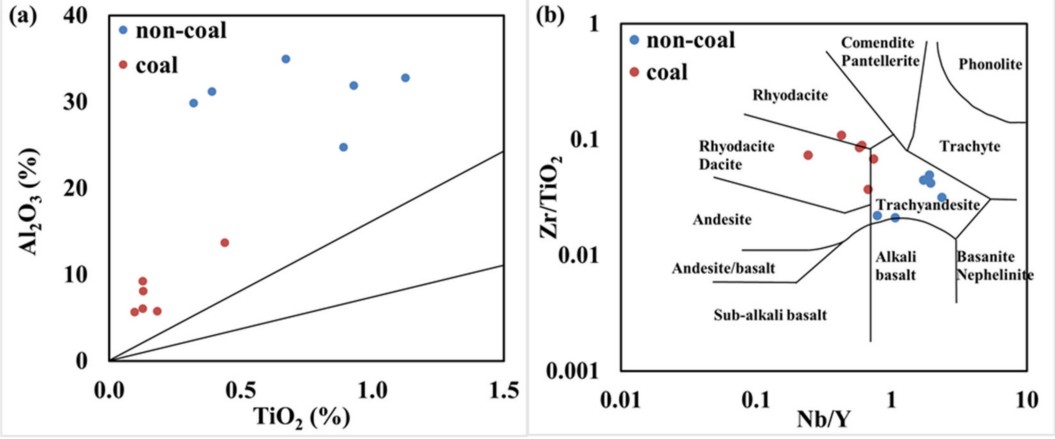

**Figure 8.** Figure of relationship between elements in Wujiawan coal. (**a**) Relationship between $Al_2O_3$ and $TiO_2$. The lower and upper diagonal lines represent $Al_2O_3$/$TiO_2$ = 8 and $Al_2O_3$/$TiO_2$ = 21, respectively. (**b**) Figure of the ratio of Zr/$TiO_2$ verus Nb/Y.

The figure of Zr/$TiO_2$ verus Nb/Y (see Figure 8b) for all the samples from the No. 5 coal were studied following Winchester and Floyd [80]. Most of the coal samples fall into the area of rhyodacite/dacite and rhyodacite, suggesting a felsic sediment source, whereas all the noncoal (parting and floor) samples fall into the field of trachyandesite; this is further evidence that the felsic composition dominated the sediment source region at the peat swamp.

The granite and gneiss of the Yinshan Oldland, which are specially enriched in Li, Ga, and high field strength elements (HFSEs, elements whose ions have a small radius and high charge, and therefore, high associated electric field [81]), have been suggested to be the main source of Li, Ga, and some of HFSEs in the coals from the Jungar coalfield and Ningwu coalfield, which are adjacent to Datong coalfield [10,71,82–84]. Therefore, the strong correlations between Li, Ga, and HFSEs in this study may indicate that the Yinshan Oldland is the potential source region of sediment of the coal from the Datong coalfield. This inference is further supported by Dai et al. who found that the average Li content in the middle Proterozoic K-feldspar granite of the Yinshan Oldland is 26 μg/g [10] (this is around two times higher than world hard coals). Meanwhile, Dai et al. reported that coals with felsic compositions in the source of sediment region have a high content of lithophile elements [23]. The ratio of $Al_2O_3/TiO_2$, along with the results from $Zr/TiO_2$ vs. Nb/Y, suggest that the region of the source of sediment for all the samples (coals, partings, and floor) is dominated by felsic terrigenous materials. The results provide further evidences for that the granite and geneiss of the Yinshan Oldland might be the source of sediment of the coal from the Datong coalfield.

Dai et al. indicated that Eu anomalies normally originate from the source region [68]. The felsic and felsic-intermediate detrital materials could result in negative Eu anomalies in the coals, while positive Eu anomalies could result from high-temperature solutions and alkali mafic rocks from source region. Apart from coal sample WJW-10 ($Eu_N/Eu_N^* = 1.08$) and parting sample WJW-11 ($Eu_N/Eu_N^* = 1.25$), the values of $Eu_N/Eu_N^*$ for all the remaining samples are lower than 1 (Table 4). Along with the ratio of $Al_2O_3/TiO_2$, the results of the figure of $Zr/TiO_2$ vs. Nb/Y, and the enrichment of the lithophile elements in the No. 5 coal, it can be inferred that most samples probably have felsic detrital input. The weakly positive Eu anomalies of samples WJW-10 and WJW-11 are probably attributed to the high content of feldspar minerals in the granite from the source region [75]. The same situation (the majority of samples displaying negative Eu anomalies while only a few individual samples have positive Eu anomalies) has been found in the Hunchun coalfield [75], as well as in the Tongjialiang mine, Datong coalfield [36]. Both studies indicated that the feldspar of granite, especially plagioclase, is the main carrier of Eu. Previous studies by Jian et al. [85] and Zhao et al. [19] showed old granites from the Yinshan Oldland that have L-type REY enrichment and a weak positive Eu anomaly. All these evidences further support that the Yinshan Oldland is the potential source region of sediment of the Wujiawan coals.

Gadolinium anomalies in coal are mainly controlled by the source regions, seawater, hydrothermal solutions, etc. [68] The Wujiawan coals have a weakly positive Gd anomalies (with a mean of 1.10) (Table 5), and a previous investigation by Kevin and Zhou suggest that a positive Gd anomaly may evidence influence of cold acidic waters [86]. However, relative high sulfur content in sample WJW-10 along with the cell-filling calcites (see Figure 4c), fracture-filling calcites (see Figure 4d), and the pyrite co-occurring with calcite (see Figure 4e), as well as H-type REY enrichment of most the coal samples suggest that the No. 5 coal may be influenced by alkaline waters (seawater or derivatives) rather than a cold acidic water contribution. All these were further supported by the marine-terrigenous depositional environment of C–P coal from Datong coalfield.

*5.3. Modes of Occurrence of Enriched Valuable Elements*

Figure 9 shows the concentration variations between ash yield and enriched elements (Li, Ga, and HFSEs) in the No. 5 coals from the Wujiawan mine. All the enriched elements exhibit similar changes, consistent with the variation of ash yield. However, because the REY in the No. 5 coal probably have both organic and inorganic affinities, they do not show a similar fluctuation associated with the ash yield in the profile. The source of REY in the No. 5 coal will be discussed separately with other enriched elements.

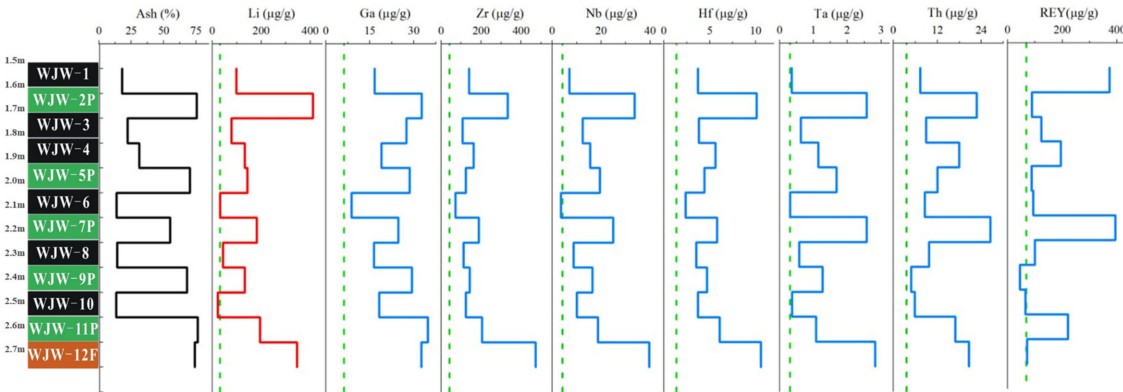

**Figure 9.** Concentration variations of ash yield and enriched elements (Li, Ga, Zr, Nb, Hf, Ta, Th, and REY) in the No. 5 Coals from the Wujiawan mine. The green dotted lines represent the average value of each element for world hard coals [65].

### 5.3.1. Lithium

The content of Li in the No. 5 coal ranged from 22.7 to 411.6 µg/g (Table 3), with a mean concentration of 67.66 µg/g, which is around seven times higher compared to world hard coals (see Figure 5) [65]. Furthermore, the concentration of Li in the partings is higher than in the coal samples, with an average of 234.88 µg/g, which is around 5 times higher than the value in world clays [66].

Due to the potential economic value, the issue of investigating the mode of occurrence of Li has become an urgent topic worldwide. Earlier studies have suggested that Li can occur both in inorganic and organic matter in coals [9,32,33,87], while other studies on the coals from the Jungar Basin indicated that Li is mainly associated with kaolinite/illite, chlorite, and boehmite [22,71,84]. However, Li mainly occurs in aluminosilicate minerals, such as kaolinite, in the coal from the Datong coalfield and Ningwu coalfield (adjacent to the Datong coalfield) [15,29]. In this study, the strong affinity of Li and ash yield (r = 0.80), along with a similar variation between Li and ash yield (see Figure 9) through the seam section, indicate an inorganic affinity of the Li in the No. 5 coal. Lithium also has strong correlation with $SiO_2$ (0.82), $Al_2O_3$ (0.78), and the lithophile elements, including Li-Zr (0.90), Li-Nb (0.93), Li-Hf (0.97), Li-Ta (0.87), and Li-Bi (0.82) (Table 5 and see Figure 10a–e), suggesting that aluminosilicate minerals probably are the main carrier of Li. The frequently observed clay minerals under the optical microscopy, XRD, and SEM-EDS provide further evidence that kaolinite (see Figures 3 and 4a,b) might be the main Li-carrier mineral in this coal. This is consistent with the results from the No. 5 coal from the Yanzishan mine, Datong coalfield, which is adjacent to the Wujiawan mine [29]. The high content of Li and strong correlations between Li and lithophile elements also reveal that the Li in the Wujiawan coal originates from the source region—Yinshan Oldland.

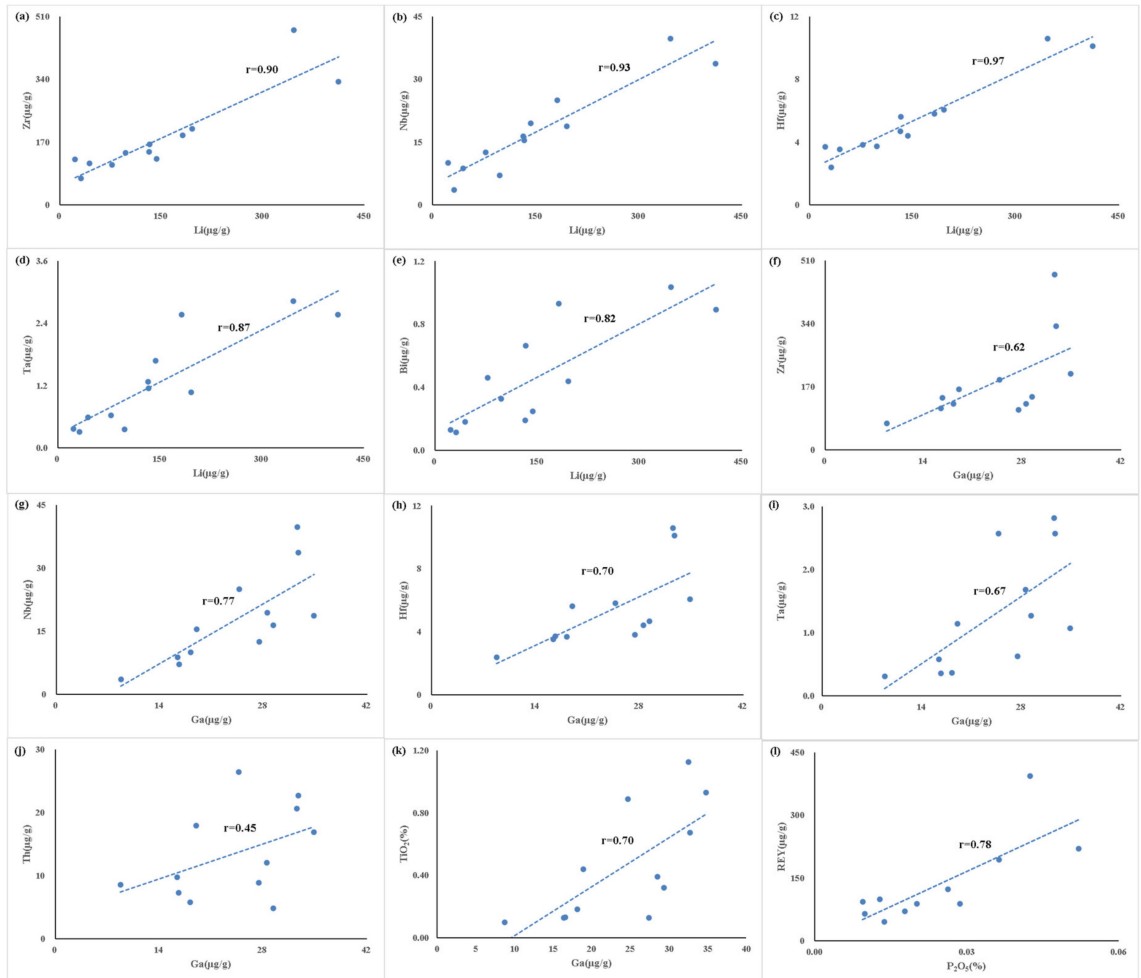

**Figure 10.** Correlation of: (**a**), Zr to Li; (**b**), Nb to Li; (**c**), Hf to Li; (**d**), Ta and Li; (**e**), Bi to Li; (**f**), Zr to Ga; (**g**), Nb to Ga; (**h**), Hf to Ga; (**i**), Ta to Ga; (**j**), Th to Ga; (**k**), TiO$_2$ to Ga; and (**l**), REY to P$_2$O$_5$.

### 5.3.2. Gallium

The content of Ga in the Wujiawan coal samples ranges from 8.7 to 27.4 µg/g and varies from 24.7 to 34.8 µg/g in the parting samples, with mean concentrations of 17.72 µg/g and 30.02 µg/g (Table 3), respectively. The average Ga content in the coal samples is around three times higher than the average Ga content in world bituminous coals (5.8 µg/g) [65], while the mean concentration of Ga is around two times higher compared to the average content of Ga in the world clays (16 µg/g) [66].

Gallium is a valuable and rare metal in coal. Ga generally has a strong affinity with the clay minerals in coal, such as kaolinite and illite [10,21,88,89]. However, Ga can also associate with other minerals, including sulfide minerals [88], goyazite [22], and svanbergite [71], etc. and organic matters [67]. In the present study, it can be inferred that inorganic matter is the main carrier of Ga, owing to its strong positive affinity with ash yield (r = 0.88, see Figure 7b). In addition, the Ga in the No. 5 coal has strong and similar correlations with Al$_2$O$_3$ and SiO$_2$ (r = 0.88 and 0.87, respectively) (Table 5), suggesting that the Ga in the Wujiawan coal mainly associate with aluminosilicate minerals (e.g., kaolinite). Furthermore, Ga is highly related to Ti (0.70) (see Figure 10k and Table 5), implying that Ga also occurs in the Ti-bearing minerals, as discussed above, Ti-bearing kaolinite may be also the carrier of Ga. A previous study by Liu et al. suggesting that aluminosilicate minerals are the main Ga carrier in the coal from Ningwu coalfield (which is adjacent to Datong coalfield) provides further support for the results [15]. The similar variation between Ga and Li, along with the high affinity between Ga and Zr, Ga and Nb, Ga and Hf, Ga and Ta, and Ga and Th (see Figure 10f–j) indicate

that Ga has the same origin as these elements and may originate from the granite and gneiss of the Yinshan Oldland.

### 5.3.3. Zirconium, Niobium, Hafnium, Tantalum, and Thorium

Zirconium, Nb, Hf, Ta, and Th are regarded as HFSEs and have been found enriched in the coals from the southwest of China, such as Chongqing [90] and Yunnan [91]. An earlier study has also revealed that HFSEs have a higher content in the coals from the Daqingshan coalfield in North China than in world hard coals [19]. Furthermore, these studies suggested that most of the HFSEs occur as absorbed ions in the clay minerals in coal. However, the organic matter can also be the carrier of HFSEs [5,22].

The average content of Zr, Nb, Hf, Ta, and Th in the No. 5 coal samples are 119.8, 9.6, 3.8, 0.6, and 9.7 µg/g, respectively. The average concentration of all HFSEs is more than two times higher compared to the mean content in world hard coals. Interestingly, the HFSEs content in the samples WJW-3 and WJW-4 (both are coal samples), which are interlayered between the parting samples WJW-2P and WJW-5P, is higher than the other coal bench samples. This may be due to the influence of groundwater when HFSEs got leached from the adjacent parting and became incorporated into the organic matter [22].

The strong correlation coefficients between HFSEs (Table 5), along with the similar variation between these elements (see Figure 9), indicate that the HFSEs probably have similar modes of occurrence. Figure 7b shows that HFSEs in the No. 5 coal have high correlation coefficients with ash, and this, together with highly positive correlation coefficients between HFSEs and $Al_2O_3$ ($SiO_2$) (Table 5), indicates that aluminosilicate minerals are the HFSEs-carrier minerals. Furthermore, the HFSEs in the No. 5 coal also have strong affinity with $TiO_2$ (Table 5), which probably suggests that the Ti-bearing clay minerals are also possible carriers for HFSEs. Considering the high correlation coefficients of Li, Ga, and HFSEs, it appears that the HFSEs also originated from the granite and gneiss of the Yinshan Oldland.

### 5.3.4. REY

Numerous investigations have suggested that REY in coal generally have a strong affinity with the ash yield and are generally associated with minerals [5,17,21,87,92], including clay minerals, phosphates, and, to a lesser extent, associated with the organic compounds in coal [4,10,21,93]. In the present study, the weak negative correlations between REY and ash (see Figure 7b) (r = −0.096), REY and $Al_2O_3$ (r = −0.095), and REY and $SiO_2$ (r = −0.075) (Table 5) suggest that REY have association with either organic or inorganic matter in the Wujiawan coal. However, REY show a strong affinity with $P_2O_5$ (0.78, see Figure 10), along with the weakly negative affinity between $P_2O_5$ and ash (see Figure 7b), which suggests that REY probably associate with organophosphates. This is supported by no P-bearing mineral being detected in the Wujiawan coal. Although the phosphorus in coal normally occurs as inorganic mineral components [94], the possibility of the presence of organic phosphorus compounds has been noted by Swaine [95].

### 5.4. Evaluation of Li in the Wujiawan Coals and Comparison with Adjacent Coals in the Datong Coalfield

The No. 5 coal from the Wujiawan mine is a type of coal, with several enriched valuable elements, such as Li, Ga, HFSEs, and REY (Table 3, see Figure 4). In particular, the content of Li is around seven times higher when compared to the mean concentration of Li in world hard coals. Earlier studies on the concentrations of Li in coals from different coal mines of Datong coalfield have been discussed in detail [29,35,36,39,96,97]. The content of Li in the Datong coalfield ranges from 13.3 to 294.6 µg/g, with a mean of 84.6 µg/g. Figure 11 shows that the Li content in coals is relatively high in the north Datong coalfield, which may be due to the area being closer to the source region, while the high Li-content area in the south Datong coalfield may be due to the influence of seawater [11]. As Sun et al. [98] reported, the minimum mineable grade for Li in Chinese coal should be set at 120 µg/g. The content of Li in the

No. 5 coal from the Wujiawan mine (67.66 µg/g) does not reach this level; however, in the northern and eastern part of the Datong coalfield there are several regions where the Li content is higher than the mineable grade. For example, in the northern Datong coalfield, there is a mine with the Li content of 294.6 µg/g (see Figure 11). This is significantly higher than the mineable grade; therefore, there is the potential for economically viable recovery of Li in these parts of the Datong coalfield. Furthermore, because of the strong affinity with kaolinite, Li in Wujiawan coal can be further enriched in byproducts of coals since Li is not considered to be the volatile element [99,100]. Thus, the coal ash, including the fly and bottom ash, should be regarded as an economically viable source for Li recovery. Considering that the coal consumption in China will continue to grow due to its rapid economic development, if the Li in all these coals occurs in inorganic associations, or partly occurs in inorganic associations, then it should be beneficial to pay more attention to the utilization process. Thus, there would be significant economic value in the recycling and utilization of Li in the fly ash of coal from the Datong coalfield, especially from the northern region of the Datong coalfield.

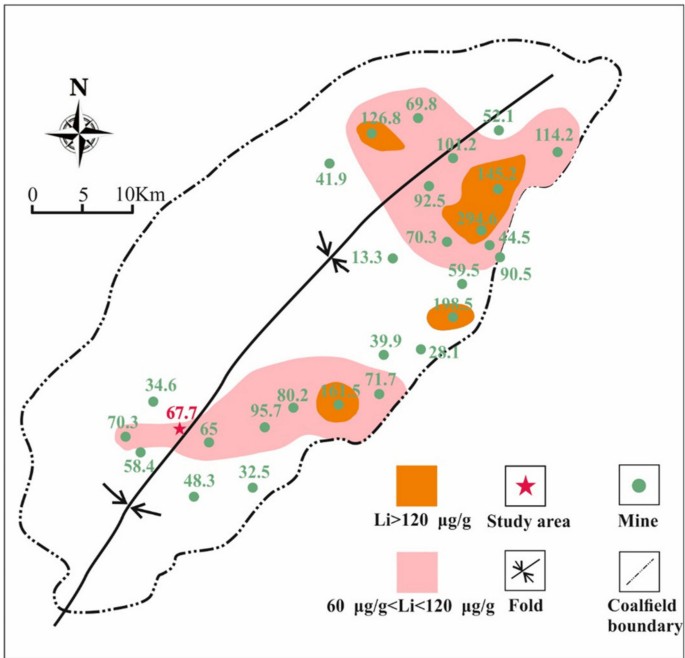

**Figure 11.** Content map of lithium in the Datong coalfield (the green number represents the concentration of Li (µg/g), data from Yuan et al. [29,35,36], Liu et al. [39], Ma [96], and Shao [97].

## 6. Conclusions

The Wujiawan coal is a kind of low-ash, medium–high-volatile, low moisture, medium-sulfur, bituminous coal. The main mineralogical compositions of Wujiawan coals are primarily characterized by kaolinite, calcite, and pyrite, along with a small amount of quartz and illite. The $SiO_2/Al_2O_3$ ratio of the No. 5 coal is very close to the value of theoretical kaolinite; this is due to the low content of quartz in the No. 5 coal seam and Si mainly occur in kaolinite. In addition, the Wujiawan coals are slightly enriched in $P_2O_5$ ( $TiO_2$, $Fe_2O_3$, $MgO$, $K_2O$, and $Na_2O$) have values lower than the corresponding average values in Chinese coals.

The Wujiawan coals are enriched in Li, Ga, HFSEs, and REY; this is particularly true for the case of Li (average concentration of 67.66 µg/g), which is around 7 times higher compared to world hard coals. Lithium, Ga, Zr, Nb, Hf, Ta, and Th have strong inorganic affinities, whereas REY have organic affinities. The main carriers of Li, Ga, and HFSEs are aluminosilicate minerals, while REY probably associate with organophosphates. The HFSEs are enriched in both partings and the adjacent coal samples, suggesting that the partings were subjected to leaching during the diagenesis. The distribution

patterns of REY, along with the ratio of $Al_2O_3/TiO_2$ and the figure of $Zr/TiO_2$ versus $Nb/Y$, suggest that these elements in No. 5 coal originate mainly from the source region with felsic detrital materials. The Yinshan Oldland may be the potential source region of sediment of the Wujiawan coals. The mean content of Li in the Datong coalfield is 84.6 µg/g. However, the Li content in several regions are higher than the mineable grade, moreover, in the northern Datong coalfield, there is a mine with the Li content of 294.6 µg/g. This is significantly higher than the mineable grade; therefore, there is the potential for economic recovery of Li in these parts of Datong coalfield. Furthermore, Li in Wujiawan coal can be further enriched in byproducts of coals. It should, therefore, be important to pay more attention to the utilization process, since there would be a potential economic value for the recycling and utilization of Li in fly ash, in particular from the northern Datong coalfield.

**Author Contributions:** J.M.: Conceptualization, Writing-Original draft preparation. L.X.: Formal analysis, Data curation. K.Z.: Formal analysis, Data curation. Y.J.: Formal analysis, Data curation. Z.W.: Formal analysis, Data curation. J.L.: Formal analysis. W.G.: Writing-Reviewing and Editing, Supervision. P.G.: methodology. S.Q.: Data curation. C.Z.: Writing-Reviewing and Editing, Supervision. All authors have read and agreed to the published version of the manuscript.

**Funding:** This study was supported by the National Natural Science Foundation of China (No. 41872173) and Natural Science Foundation of Hebei Province (No. 2016402104).

**Acknowledgments:** Bangjun Liu is thanked for his constructive suggestions and language checking for this manuscript. The authors sincerely thank Brigitte Held from the GRADE of Goethe University for the assistance of language polishing.

**Conflicts of Interest:** The authors declare no conflict of interest.

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
