# Peer review of "Geochemistry of Carboniferous–Permian Coal from the Wujiawan Mine, Datong Coalfield, Northern China: Modes of Occurrence, Origin of Valuable Trace Elements, and Potential Industrial Utilization"

_minerals, doi:10.3390/min10090776_

Round 1

Reviewer 1 Report

The manuscript by Ma et al. describes geochemistry and mineralogy of one of the Upper Paleozoic coal seams in North China. This topic fits the journal scope. The paper is well organized, including high-quality figures and tables with the data obtained. Unfortunately, some conclusions of this work are not solid. Causes of the latter seem derived from some gaps in the data. For example, the presented mineralogical data include just a single XRD pattern of one coal sample and several SEM images that do not allow correlation between chemical and mineralogical compositions of the studied rocks. The geochemical data lack concentrations of some trace elements. For instance, there is no data for Sr and B, which may be used to estimate a possible influence of seawater on some element enrichments discussed in the text.

As a result, no strong evidence is present to substantiate the supposed hydrothermal influence. High sulfur content in WJW-10 and presence of calcite and pyrite may be attributed to influence of seawater and/or its derivatives.

The Yinshan Upland is a highly probable source of the studied sedimentary rocks, since it is close to the Datong Coalfield geographically. However this suggestion has not been confirmed by the data from this study, while referencing of the previous works is invalid. Felsic rocks are widespread in North China as in any cratonic part of the Earth.

REY and HFSE are described in the paper as separate groups of elements, but actually REY are a part of HFSE.

More specific comments may be found in the attached pdf file.

Reviewer 2 Report

This is a well organized paper with a rich set of geochemistry data on the coals. The authors investigated the minerals and trace elements of the Carboniferous-Permian coal from the Wujiawan Mine, Datong Coalfiled, Northern China. Correlation analysis has been carried out to study the affinity and modes of occurrence of the elements. The high elevated Lithium in the study coal and parting samples made the coal ash possible to be an economically viable source for Li recovery. I recommend acceptance for publication after a major revision.

Line 110 Please add Moisture as the authors cited the Ref.31 D3173-11 in the next line.

Line 111 Unfortunately the Ref.34 is the standard method for forms of sulfur. Please cite the references properly.

Line 127-129 The Ref.26 and 36 presented two different digestion procedures, please give a description of the microwave digestion procedure in the present stud. Again, please cite the references properly and correctly.

Line 149 “as-received basis” is generally abbreviated with “ar”.

Line 172 The cell-filling calcite indicates it is of anthigenic origin rather than epigenic origin.

This is inconsistent with the cited references here.

Line 186 Fig.4b It is strange the EDS data shows no Au, as each sample for the SEM was gold-plate coated as the authors have described in the “Sampling and Methods” section Line 118-119. Make an explanation somewhere in the paper will be necessary.

Line 194 The term “Chinese hard coals” is improper here. Not only “hard coals”, but also “brown coals” actually have been calculated in the study by Dai et al. (2012).

Line 199 The major element oxides of the coal ash, determined by XRF, certainly include the Si occurred in the kaolinite.

Line 247-249 The anomalies of Sm, Dy, He and Tm in the coal samples are unusual, please use them in caution and double check the REE concentration data. Retest the concentration of REE is recommended.

Line 271-274 “…the TiO2 … has strong and similar correlations with Al2O3 and SiO2…this is also supported by the low content of titanium oxide minerals…” this statement is illogical, please rephrase it.

Line 353-354 Figure 9, “Ashe” should be changed to “Ash”

Line 388 remove “associations”

Round 2

Reviewer 1 Report

I am almost fully satisfied by the authors’ revision. However some minor problems remain:

  1. No evidence is present that the Yinshan old granites and gneisses are rich in Li and associated elements. This disadvantage is inherited from the previous coal studies. However, this work may be of higher value, if the authors have found and reference the appropriate study.
  2. I am not an expert in English, but can see that the manuscript needs some language correction and polishing.

Reviewer 2 Report

The new submitted manuscript has been greatly improved with the reviewers’ suggestions and comments. The authors’ responses and the revised manuscript are acceptable in general. 
